# Laser Interference Lithography—A Method for the Fabrication of Controlled Periodic Structures

**DOI:** 10.3390/nano13121818

**Published:** 2023-06-07

**Authors:** Ri Liu, Liang Cao, Dongdong Liu, Lu Wang, Sadaf Saeed, Zuobin Wang

**Affiliations:** 1International Research Centre for Nano Handling and Manufacturing of China, Changchun University of Science and Technology, Changchun 130022, China; liuri1994@163.com (R.L.); caoliang@cust.edu.cn (L.C.); liudong@mails.cust.edu.cn (D.L.); wangl@cust.edu.cn (L.W.); sadaf0635@gmail.com (S.S.); 2Centre for Opto/Bio-Nano Measurement and Manufacturing, Zhongshan Institute, Changchun University of Science and Technology, Zhongshan 528437, China; 3Ministry of Education Key Laboratory for Cross-Scale Micro and Nano Manufacturing, Changchun University of Science and Technology, Changchun 130022, China; 4JR3CN & IRAC, University of Bedfordshire, Luton LU1 3JU, UK

**Keywords:** laser interference lithography, laser materials processing, periodic structure, micro/nanostructuring, surface functionalization

## Abstract

A microstructure determines macro functionality. A controlled periodic structure gives the surface specific functions such as controlled structural color, wettability, anti-icing/frosting, friction reduction, and hardness enhancement. Currently, there are a variety of controllable periodic structures that can be produced. Laser interference lithography (LIL) is a technique that allows for the simple, flexible, and rapid fabrication of high-resolution periodic structures over large areas without the use of masks. Different interference conditions can produce a wide range of light fields. When an LIL system is used to expose the substrate, a variety of periodic textured structures, such as periodic nanoparticles, dot arrays, hole arrays, and stripes, can be produced. The LIL technique can be used not only on flat substrates, but also on curved or partially curved substrates, taking advantage of the large depth of focus. This paper reviews the principles of LIL and discusses how the parameters, such as spatial angle, angle of incidence, wavelength, and polarization state, affect the interference light field. Applications of LIL for functional surface fabrication, such as anti-reflection, controlled structural color, surface-enhanced Raman scattering (SERS), friction reduction, superhydrophobicity, and biocellular modulation, are also presented. Finally, we present some of the challenges and problems in LIL and its applications.

## 1. Introduction

Nature is full of strange creatures, and many people have the impression that certain unique micro/nanosurfaces in nature are non-regular structures, but this is not the case [1]. For example, the beautiful seashells, the wings of butterflies, the feathers of birds, the scales of pangolins, and the epidermis of sharks are all periodic-ordered structures [2,3,4]. Furthermore, it is these microscopic periodic-ordered structures that give them various macroscopic properties such as wettability, wind/water resistance reduction, friction reduction, reflection reduction, and different structural colors.

As these regular surfaces are studied more thoroughly, it has been found that objects work better when these structures are “imitated” in a variety of ways on the surfaces [5,6,7,8]. In recent decades, periodic regular structures have received increasing attention in many fields, and the study of these rules has become increasingly sophisticated. For example, researchers have studied wettability, and people prepare certain specific surface structures that can achieve the phenomenon of water flowing to high places [9,10,11]. Some swimmers wear special shark suits and their speeds improve considerably [12,13]. There are so many examples of these periodic regular structures benefiting human life. It is possible that a random object that people own contains a micro/nano-periodic structure that was designed by researchers.

Currently, as many as dozens of methods are used to produce periodic regular structures. From the broad categories, they can be divided into mechanical, optical, and chemical methods [6,14,15]. Mechanical methods include scribing, EDM, nanoimprinting, water flow, etc.; chemical methods include electrochemical, hydrothermal, CVD deposition, etc.; and optical methods include femtosecond lasers, direct laser scanning, laser interference lithography (LIL), etc. However, these techniques have limitations in terms of low yield, small patterning area, and/or high equipment, and tooling cost. In contrast, LIL works on large areas at a low cost [16,17,18]. Periodic structures are of interest for their inherent merits in functional applications such as diffraction modulation, controllable wettability, and reflection reduction, as well as for their different structural colors [19]. There are many techniques capable of fabricating periodic structures, among which the LIL technique has incomparable advantages for the fabrication of periodically large-area patterned structures [16,20,21,22].

LIL is an advanced micro- and/or nanoprocessing technique that can be used to produce high-resolution micro- and/or nanostructures and devices [23]. LIL uses the interference property of light to realize multiple coherent laser beams that meet on the substrate surface to form a bright and dark interference region with periodic light intensity distribution. The traditional lithography process is to expose the sample and transfer the pattern to the substrate surface through processes, such as development, etching, or coating, to form a patterned substrate. The periodic intensity distributions in the interference region are ‘recorded’ on the substrate, so LIL is a maskless optical exposure technique that avoids the use of mask plates and reduces the cost of lithography. LIL can produce large-area, high-resolution periodic patterns at a low cost, while the exposure process is simple and the pattern period can be flexibly changed [16]. LIL can produce patterns not only on flat substrates, but also on curved or partially curved substrates using its large depth of focus.

This review describes the basic principles of LIL and its applications in various fields. Section 2 describes the differences between LIL types in terms of beam number and number of exposures, and discusses the effects of spatial angle, angle of incidence, wavelength, and polarization state on the laser interference light field. Section 3 also reports on the diverse periodic structures that can be prepared using LIL for applications in functional surfaces such as anti-reflection, control of structure color, surface enhanced Raman scattering (SERS), friction reduction, superhydrophobicity, and biological cell modulation. Finally, we present some of the challenges and problems that LIL has encountered in not achieving full industrialization.

## 2. Laser Interference Lithography

### 2.1. Types of Laser Interference Lithography

Laser interference lithography can be classified into various forms. According to the number of beams of interference, LIL can be divided into double-beam, triple-beam, quadruple-beam, quintuple-beam, and other multi-beam interference lithography [16]; by the number of exposures, LIL can be divided into single-exposure lithography and multi-exposure lithography [24]. Other lithography systems currently have to shorten the wavelength and sacrifice the depth of focus in order to pursue high resolution. Ion beam lithography and electron beam lithography have sufficient resolution, but their slow fabrication process limits their application in high-volume manufacturing. LIL is a fine graphics processing technique with a low cost, simple system, and high resolution. It has great advantages and a key position in preparing array structures. Its system is inexpensive, does not require complex optical components, it also has expensive imaging lenses and is capable of preparing graphics in large areas without the depth-of-focus limitation. It has a high resolution and does not require masks, making it a promising lithography technique at present.

LIL exploits the interference properties of light to realize multiple coherent laser beams that meet on the substrate surface to form bright and dark interference regions. The intensity distributions in interference regions are characterized by periodicity [16]. LIL can produce controlled periodic or quasi-periodic structures on nano- and microscales, and fine periodic array structures can be fabricated under the strict control of the process and exposure dose [25,26,27,28,29,30,31].

Micro/nanostructures with different feature sizes are obtained through LIL. The interference of three laser beams is considered as the superposition of their electric field vectors. It can be expressed as [32]:(1)$$\vec{E}=\sum_{n=1}^{3}{\vec{E}}_{n}=\sum_{n=1}^{3}A_{n}{\vec{p}}_{n}\cos\left[k{\vec{n}}_{n^{\cdot }}{\vec{\gamma }}_{n}\pm 2\pi \nu t+\phi_{n}\right]$$
where $A_{n}$ is the amplitude of the electric field vector and ${\vec{p}}_{n}$ is the unit polarization vector. k=2π/λ is the wave number (λ = wavelength). $\vec{n_{n}}$ is the unit propagation vector in the wave propagating direction, ${\vec{\gamma }}_{n}$ is the position vector, $\phi_{n}$ is the phase constant, and ν is the frequency. A flexible LIL system can select the parameters and obtain the designed interference pattern.

The interference intensity is expressed as [26]:(2)$$I={\left|\vec{E}\right|}^{2}=\sum_{n=1}^{3}\sum_{m=1}^{3}\left|{\vec{E}}_{n}\right|\left|{\vec{E}}_{m}\right|\cos\left\langle \vec{E_{n}}\cdot {\vec{E}}_{m}\right\rangle$$

### 2.2. Influence of the Process Parameters for Laser Interference Lithography

Laser interferometric lithography produces controllable periodic or quasi-periodic structures from nanoscale to microscale, producing fine periodic array structures with a tightly controlled process and exposure dose [22]. Several factors determine the interference pattern and related parameters:The number of laser beams producing different micro- and nanoarray structures;The incident angle preparing the grating structure of different periods;The azimuth angle producing different morphology arrays;The energy of the laser determining the depth of the structure; andThe polarization state determining the pattern profile.

The basic principle of laser interference lithography is to combine two or more laser beams that meet the interference conditions according to the interference principle of light and change the exposure time, phase angle, incident angle, and other parameters that can obtain different periodic pattern arrays such as stripe array, dot array, and dimple (hole) array.

#### 2.2.1. The Number of Laser Beams

The different numbers of laser beams produce different structures such as streak arrays, dot arrays, and dimple (hole) arrays. As shown in Figure 1, Ainara Rodriguez et al. [16] used MATLAB software to perform laser interference simulations to obtain schematic diagrams of the optical field distribution for different interference conditions. Figure 1A,B show double-beam interference simulations, where stripe arrays can be obtained under interference conditions; Figure 1C–E show triple-beam interference light field simulations, where a hole array structure (Figure 1C) or dot arrays (Figure 1D,E) can be obtained under triple-beam conditions; Figure 1F–J show quadruple-beam interference light fields. F and G are dot/hole arrays produced by four beams with different incidence angles. (Figure 1H–J) are arrays of dots/holes produced by four beams with different angular positions.

The repetition distance between the maximum and minimum values of the interfering intensity, which is called the spatial period (*P*), basically depends on the incidence angle *θ* between the interfering beams and the laser wavelength *λ*. In the case of double- and triple-beam interferences, the spatial period is described by Equations (3) and (4), respectively [30,33].
(3)$$P=\frac{\lambda }{2\;sin\;\theta }$$
(4)$$P=\frac{\lambda }{\sqrt{3}\;sin\;\theta }$$

#### 2.2.2. The Azimuthal Angles

The effect of azimuth angles on the interference of double-, triple-, and quadruple-beam lasers has been studied by Jinjin Zhang et al. [27]. The modulation mode is not formed at an equal azimuthal angle for triple-beam interference. It is a regular triangle interference pattern distribution of holes or dots, while the azimuthal angles are symmetrical. The distance between two holes or dots in the pattern is $4\frac{P_{0}}{3}$. However, the interference distribution is not a regular triangle in most situations. The interference patterns are based on the geometric distribution of an array of holes or dots. The angle value is half the difference between any two azimuthal angles as the interference pattern can be considered as the superposition of the three interfering beams, as described by Equation (2). The energy is redistributed with the change of azimuthal angles. The holes or dots will be elliptical, as the azimuthal angles are not centrally symmetric, and become slender in the pattern.

The modulation period can be calculated by [27]:(5)$$P=\frac{\lambda }{2\;sin\;\theta \cdot sin\frac{\varphi }{2}}=P_{0}\cdot \frac{1}{sin\frac{\varphi }{2}}$$
where $P_{0}$ is the original period and *φ* is the azimuth angle. In quadruple-beam interference, the azimuth angles change with the modulation period. According to Equation (3), the pattern period in Equation (5) is increased by $\frac{1}{sin\frac{\varphi }{2}}$*,* compared to the original period $P_{0}$. It can be seen that the variation of the azimuthal angles can cause a change in the modulation period. In the TE-TM-TE-TM polarization state, the modulation period increases with the decrease in the azimuth angles. In quadruple-beam interference, the azimuth angle changes with the modulation period.

#### 2.2.3. The Energy of the Laser

The laser energy determines the depth of the structure. Dapeng Wang et al. [25] used the AFM to measure the fluctuation change of the interference texture under different energy conditions. Figure 2 shows the stripe structures of double-beam interference, which are from single-pulse exposures on the silicon surface at total energies of 590 and 630/700 mJ·cm^−2^, respectively. The effect of energy on the structural contrast can be clearly seen from the AFM map, and the higher the energy, the higher the contrast. Through the rational selection of process parameters, nanosecond laser interference lithography is ideal for the flexible fabrication of micro/nanostructures for various applications such as anti-reflection and self-cleaning surfaces.

#### 2.2.4. The Polarization State

Apart from the above parameters, the effect of the polarization state on the pattern formation was investigated. Jianfang He et al. [29] discussed the relevant research by placing a half-wave plate at the input coupling or in front of the binding end of three fibers in triple-beam interference. It is a rectangular triangle interference distribution pattern and the azimuthal angles are 0°, 90°, 270°, as shown in Figure 3a. The polarization direction of the initial vertically polarized 325 nm laser beam was varied from α = 0 to 90 degrees in steps of 10 degrees. Where α is the angle of the in-plane polarization direction relative to the vertical axis perpendicular to the axis of the fiber beam. Figure 3b shows the simulation results of the interference pattern within the 22 μm^2^ region, and Figure 3c shows the experimental results of the corresponding photoresist grating, with an area of 55 μm^2^. It can be seen that by increasing the value of α, the dot units become slender and finally form grating in the pattern. Therefore, it was found that the polarization not only affects the structure contrasts, but also changes the interference patterns in this case.

Polarization affects the formation of interference pattern, pattern contrast, and period in quadruple-beam interference lithography. Dapeng Wang et al. [26] proposed three different polarization modes through theoretical analysis, simulation, and experiment for studying the effect of polarization on quadruple-beam laser interference. (Figure 4) It was found that the secondary periodicity or modulation was due to the misaligned or unequal incident angles in the TE-TE-TM-TM mode.

In short, this section distinguishes the types of LIL with the number of laser beams and the number of exposures of the laser. Furthermore, it focuses on how factors, such as spatial angle, incidence angle, wavelength, and polarization, affect the laser interference light field, and as well as their laws.

## 3. Applications of LIL

In the previous section, we introduced the LIL technique and its main principles. This section is focused on LIL applications in the fields of solar panels with anti-reflective properties, controlled structural colors and photonic crystals, nanoparticle preparation with catalytic properties, SERS, biomedicine, and functional surfaces (superhydrophobicity, anti-icing/frost properties, friction reduction, and hardness enhancement).

### 3.1. Solar Panels with Anti-Reflection Properties

The impending depletion of fossil fuels poses a major energy challenge to humankind, while using fossil fuels causes serious environmental pollution and contributes to the greenhouse effect. According to the UN Millennium Plan, climate change is a major challenge for the world. Scientists predict that the continued global warming will have serious environmental and economic consequences for rainfall, drought, crop growth, river flows, melting ice sheets, and rising sea levels. To deal with these challenges, the development of new renewable energy sources is currently the focus of research by scientists in many countries. Among the wide range of renewable energy sources available, solar photovoltaic technology is considered to be one of the cleanest and safest large-scale power generation technologies [34,35,36].

In 1839, the French physicist Becquerel first discovered the “photovoltaic effect”, and scientists focused their research on the mechanism of the photovoltaic phenomenon and the exploration of photovoltaic materials [36]. Since then, photovoltaic power generation has become an important way of using solar energy. In recent decades, the rapid development of photovoltaic industrialization has made photovoltaic power generation a major competitor in energy production, supply, and consumption, and currently there are a variety of solar cells that can be classified in terms of the materials used [35,37]: silicon-based thin-film, dye-sensitized, organic, and quantum dot solar cells.

In solar power systems, photovoltaic modules, often called solar panels, are the core part of the systems. Their function is to convert the sun’s radiant power into electrical energy, either to be sent to batteries for storage or to drive loads. The wavelength excited by sunlight depends on the type of semiconductor. The factors that influence the efficiency of solar panel power generation are: solar irradiation intensity, module temperature, installation angle, and ambient temperature [37,38]. It has been a challenge to improve the solar cell efficiency of these solar cells without the influence of external factors. Solar cells are excited by sunlight, which causes electron transfer to generate electricity. However, the issue is how to increase the transfer efficiency of a solar panel. Scientists have devised a number of methods, one of which has a high level of acceptance. By using the “light trap effect” of the microstructure to reduce the reflection of sunlight, more light energy is retained on the surface of the panel, thus increasing the transfer efficiency of the panel.

Destructive interference, light trapping, and gradient refractive index principles are all generally accepted mechanisms for structural reflection reduction [39]. For conventional transparency, enhancing films widely used in optical and optoelectronic systems, monolayer dielectric films (n, and n < ns) with low refractive index are prepared on a substrate with high refractive index (ns) based on the thin film interference law (Figure 5a). Many natural photonic structures are not of the thin film type and have a distinct periodic variation [40,41]. Such functional structures behave differently due to their different scales. If the structural unit scale is much larger than the incidence wavelength of the light, then when the incident light enters between the macroscopic structures, part of the light is absorbed and the remaining part is usually reflected and scattered. If the structure scale is larger than the incidence wavelength of light, the incident light will be trapped in the structure gap and continuously reflected, creating a trapped light effect (as shown in Figure 5b). Conversely, if the structure size is smaller than the incidence wavelength of the light, the gap between the structure and the surrounding air medium forms a gradient refractive index layer, similar to a multilayer gradient medium film, and the incident light gradually ‘bends’ into the substrate as the refractive index changes (Figure 5c,d). This is the basic principle of the gradient refractive index, also known as a subwavelength structure. Even if the angle of incidence of light is changed, the subwavelength structure coating has a gradient effective refractive index in that direction of incident light.

LIL has the characteristics of high resolution, high speed, low cost, large area, and is suitable for mass production. For the structure shown in Figure 5b, LIL can easily and quickly produce such a high-contrast periodic structure. C. E. NEBEL et al. [42]. proposed the idea of “submicron silicon structures for thin-film solar cells” in 1995. Subsequently, laser interference has been continuously investigated for the fabrication of “trapped light effect” structures with high absorption and reflection reduction effects. In the past two decades, a variety of interferometric periodic structures have been fabricated on silicon-based solar panels. Litong Dong et al. [43,44,45] have designed and fabricated a moth-eye mimetic structure to achieve ultra-wide spectral reflectance reduction from visible light (400 nm) to mid-infrared (11 μm) with an average reflectance better than 2%, and on this basis, they have realized a multifunctional composite structure with superhydrophobic and anti-corrosion properties (Figure 6).

### 3.2. Tunable Structural Color and Photonic Crystal

The beautiful wings of butterflies, the wings of birds, and the discolored interior of seashells are all structural colors resulting from different microstructures. Structural color has nothing to do with pigment coloring, but is an optical effect caused by the sub-microstructure of living organisms. In other words, the different absorption and reflection scattering of different wavelengths of light by different micro/nanostructures produce different visual effects [46,47]. Inspired by different biological epidermal microstructures in nature, scientists can produce different controllable structural colors by changing different periodic structural parameters, groove parameters (peak width, peak height, and groove width), and a fixed structural surface can be viewed from different angles to adjust to different structural colors. These controllable structural colors can be prepared using various methods such as scribing, embossing, laser processing, and chemical etching. They have applications and research in the fields of anti-counterfeiting, camouflage, information encryption, filters, sensors, and smart windows. Much attention has been paid to the use of various micro/nanofabrication methods to fabricate well-designed micro/nanostructures for the preparation of structured color surfaces [48,49,50]. Laser interference has a unique advantage in the field of controlled periodic structures as a processing method that allows for fast, large-area, and one-shot fabrication.

Bogdan Voisiat et al. [47] developed a method to fabricate periodic structures with variable periods using direct LIL to improve the homogeneity of structure’s color (Figure 7). This method can be used to create groove structures directly on metal surfaces. The surface shows multi-colors with the diffraction grating effect of the periodic groove structure. This is because of its good flexibility and wide material applicability. This versatile metal surface has a wide range of applications in identification codes, decorative beautification, anti-counterfeiting, information storage, and design for bionic applications.

Photonic crystals are an important branch of structural color, and the concept of photonic crystals arose in 1987 when E. Yablonovitch and S. John independently recommended a structure in which the refractive index varies periodically in one or more directions, respectively [51,52]. Photonic crystals are defined as the structures with the dielectric constant (refractive index) varying periodically and having the ability to direct and confine electromagnetic waves [53,54]. Photonic crystals have important applications in many high-tech products, such as sensors, antennas, filters, beam splitters, and amplifiers. The main manufacturing methods for photonic crystals are silicon-on-insulator (SOI) [55,56,57], evaporation [58], sputtering [59,60,61], coatings [62,63,64,65], deposition [66], wet etching (solution etching, anodic oxidation, etc.), dry etching (reactive ion etching, plasma etching, inductively coupled plasma, etc.) [67,68], and electron-beam lithography [69,70]. However, the above methods are either costly or the components or products involved in manufacturing are extremely corrosive and hazardous. In contrast, laser interference lithography is expected to be an emerging technique for the preparation of photonic crystals due to its simplicity, low cost, and environmental friendliness. Laser interference lithography was first applied in the late 20th century for the preparation of two- and three-dimensional photonic crystals [71,72,73,74,75,76,77,78]. Subsequently, people began to study this technique intensively, and through a large number of experimental analyses and numerical simulation optimizations, the preparation of photonic crystals by interferometric lithography gradually matured.

In contrast to other techniques for preparing photonic crystals, laser interferometric lithography allows for the controlled preparation of complex three-dimensional structures to meet the stringent requirements for photoconductive or light-trapping functions. However, three-dimensional photonic crystals with complex functionality place high technical demands on interferometric systems. Perhaps the low-cost mass production of simple one- and two-dimensional photonic crystals is a favorable direction for photonic crystal applications. Saraswati Behera et al. [41] reported an interferometric lithography (IL) method based on a phase-space light modulator (SLM) to achieve hexagonally packed helical photonic structures with submicron periods over large areas (Figure 8).

### 3.3. Patterned Nanoparticles

The synthesis and characterization of nanoparticles and their special optical, electronic, magnetic, and catalytic properties are important branches in the current wave of nanoscience [79]. Metallic nanomaterials have extremely important applications in many fields such as catalysis [80], novel sensors [81,82,83], magnetic storage [84], surface enhanced Raman spectroscopy (SERS) [85], and biomedicine [86,87,88,89]. Patterned nanoparticles have received great attention for their wide ranges of applications and easy integration that can lead to enhanced optical, chemical, thermal, and magnetic properties of the devices [90]. Various methods have been developed to assemble nanoparticles into patterned nanostructures, exploiting different routes and factors that influence the self-assembly of nanoparticles.

With the development of micro- and nanofabrication technologies, methods for fabricating patterned nanoparticles on planar and structured substrates fall into five broad categories including: (1) lithography-assisted colloidal self-assembly methods [91,92], (2) chemical synthesis methods at solid–liquid interfaces [93], (3) photoinduced dielectrophoresis methods [94], (4) templated de-wetting methods [95,96], and (5) biological template self-assembly methods [97]. These methods demonstrate the diversity of ordered nanostructured surface preparation.

Although several methods have been developed for fabricating ordered nanoparticles, they usually require a relatively complex multi-step lithography process to obtain a patterned template, followed by chemical synthesis or particle self-assembly to achieve the assembly of nanoparticles [91,92]. Traditional methods either cannot achieve nanoparticle preparation in one step; or cannot finely control the particle size or spacing of nanoparticles; or have high fabrication costs and low efficiency and cannot achieve large-area, high-throughput, and highly controllable patterned nanoparticle preparation.

To precisely place the desired nanoparticles on a given area of surface is a challenge for nanofabrication technology. Neither bottom-up self-assembly nor top-down lithography can adequately address this challenge. Therefore, lithography-defined features are now commonly used to assemble nanostructures into ordered patterns [79]. LIL can be used to investigate the methods of patterned nanoparticle preparation and achieve effective control of properties, such as nanoparticle distribution, morphology, particle size, particle spacing, and structural components, to achieve the desired results (Figure 9) [33]. Typically, variation of nanoparticle morphology, particle size, and distribution have a very significant effect on various properties of nanomaterials such as optical (SERS) [98] and catalytic properties [99].

#### 3.3.1. Catalytic Performance

The catalytic role of gold nanoparticles in various fields has been widely recognized and applied [80]. However, metal nanoparticles can easily aggregate in applications, leading to a significant decrease in catalytic activities. The most effective way to prevent aggregation is to solidify metal nanoparticles on the supports of carbon materials, metal-organic frameworks, and various polymeric substrates. This criticism is well addressed by laser interference, such as metal-assisted chemical etching (MACE), which is a cost-effective method for fabricating silicon nanostructures including silicon nanowires (SiNWs) and silicon nanopores (SiNHs). However, the preparation of MACE metal templates requires complex experimental conditions including a stringent cleaning process and multiple steps.

As shown in Figure 10, Xudong Guo et al. [100,101] used superluminescence-enhanced laser interference lithography (SELIL) to directly fabricate complex metal patterns, and then used MACE to obtain hybrid SiNW and SiNH arrays. Ag films were first deposited on the Si substrate, and then a double-beam interferometric electric field was generated using a 1064 nm high-power laser source. As Ag particles are very sensitive to the change in input energy, they tend to decompose or aggregate and form different Ag patterns with a specific energy threshold to reduce their free energy. By controlling the distribution of the input optical field, complex metallic patterns and their corresponding Si nanostructures can be obtained with feature sizes ranging from tens of nanometers to a few microns.

#### 3.3.2. SERS

The optical properties of metal nanoparticles have attracted the attention of scientists since the Middle Ages. Gold nanoparticles selectively absorb visible light at specific wavelengths and exhibit colorful colors, a property that was widely used in the 17th century to make stained glass for churches. Surface-enhanced Raman scattering, since its discovery by the British scientist Fleischmann [102] in 1974, has been developed into an efficient single-molecule detection and material analysis technique after nearly 50 years of research [103]. The SERS technique has the advantages of high sensitivity, low water interference, and no damage to the sample, which makes it ideal for studying surface effects and analyzing the structure of substances. Following more than four decades of development, SERS technology has matured for use in many frontier areas such as chemical detection, nanoprobing, and life sciences [104].

As mentioned above, laser interference can prepare a layer of nanoparticles on the surface layer of the carrier, and the properties of nanoparticle distribution, morphology, particle size, particle spacing, and structural components can be effectively controlled. Huijuan Shen et al. [98,105,106] also proposed a laser interference-induced forward transfer technique (LIIFT) to transfer metal Ag particles on transparent media (Figure 11). In addition, in the period range of 10 to 17 μm, the larger the period of stripes, the smaller the average diameter, and the higher the density of Ag-NPs, which is the result of the transfer of micro-stripe distribution based on the double-beam LIL in the TM-TM mode. Finally, the SERS properties were discussed based on the RhB analytes and the transferred micro-stripes. The results show that the SERS intensity on Ag-NPs micro-stripes is larger than that on bare Si substrates, and the Raman spectral intensity increases with the increase of the micro-stripe period, which is mainly due to the local surface plasmon enhancement of Ag-NPs. Aggregation of silver nanoparticles improves with the increase of the micro-stripe period and the intensity of the excitation electric field is enhanced. Testing this result provides a new technique for the effective preparation of nanoparticles with controlled aggregation, and demonstrates the potential LIIFT application in the sensitivity detection of SERS chips.

### 3.4. Biological Field—Biological Cell Regulation

As an important tool for in vitro cell research, micro/nanostructures play an important role in cell adhesion [107], migration [108], growth [109], and differentiation [110,111,112], and are of great significance for wound healing, tissue repair, and disease prevention. The rapid development of micro- and nanomanufacturing technology has led to the intelligent and diversified preparation of biomaterials, which provides good technical support for the construction of the extracellular microenvironment. With the further understanding of the interaction between cells and their growth microenvironment, the study of the regulation of cell behavior by micro- and nanostructures has become a key direction of in vitro cell research. Investigating the cell responses to micro- and nanostructures provides a basis for the study of cellular response mechanisms in vivo. Biomaterials influence the cell-biomaterial interface interactions through their surface chemical energy, surface micro/nanomorphology, and other factors [113], which in turn affect various physiological activities of cells. LIL technology is the most commonly used technique for fabricating micro- and nanostructures due to its large area of action in the laser coherence range, short experimental cycle time, and relative simplicity of operation. Many researchers have used LIL-prepared structures to study the relationship between cells and environmental conditions.

As seen in Figure 12, Xiaomin Wu et al. [113,114,115] investigated the preparation of various micro- and nanostructures by LIL and the effect of these structures on the regulation of cell behavior. Based on LIL technology, they combined it with nanotransfer and metal-assisted chemical etching methods to obtain micro/nanostructures with rich mechanical cues in different stiffness and dimensions for regulating cell adhesion, spreading, and proliferation behaviors. The effects of their explored micro/nanostructures on cell behavior contribute to the study of cancer cell capture and local growth of single cells, and provide valuable references for the study of in vitro tissue construction and biomedical material implantation.

Qi Liu et al. [115] used LIL in Ti-6Al-4V to prepare stripe, hole array, and dot array structures, and investigated the response of human osteoblast-like osteosarcoma cells (MG63) to the surface modification of Ti-6Al-4V implanted alloy. The results showed that the laser-treated surface had a positive effect on the proliferation of osteoblasts after 24 h. Surfaces with this microstructure can influence cell activity and improve biocompatibility. As mentioned above, the surface prepared using LIL has a friction-reducing effect, so the LIL method of preparing implant surfaces is a practical study that produces implants with a good biocompatibility while having low friction and low losses. Research related to this method can contribute to the manufacturing of lifelong implants that can reduce the risks associated with implant replacement surgery as well as the economic losses.

### 3.5. Processing of Functional Surface Structural Components

As society has evolved, the need for functional surfaces has increased. Functions, such as superhydrophobicity [116,117,118,119,120,121], anti-icing [118,122,123,124], corrosion resistance [125,126,127,128], friction reduction [129,130,131], and high hardness [130,132,133,134], have been of great interest. In order to produce better engineered materials, researchers have developed new methods to build materials that resemble natural structures and functions [135,136,137,138,139]. It is an unchanging fact that structure determines performance! So the design and preparation of functional surfaces are also the design and preparation of various periodic composite structures.

#### 3.5.1. Hydrophobic and Anti-Ice/Frost Properties

Surface icing hazards pose a serious danger to industries such as electronic transportation, aviation, and shipping. In the aviation industry, for example, surface icing can make the aircraft heavier, less fuel efficient, and aerodynamically disturbed, which can compromise flight safety [61,140,141]. Therefore, anti-icing has become an important research direction, and numerous studies have found that nano- and microstructures with excellent water-repellent properties can reduce ice adhesion [142,143,144]. It is worth noting that the surfaces with optimal anti-icing effect are micro/nanostructure layered structures [143,145,146,147].

LIL has been relatively well researched in the preparation of micro- and nanostructures. The Fraunhofer Institute is one of the early research organizations in the industrialization of laser interferometric lithography, and they have developed various types of laser heads for the processing of micro- and nanostructures (Figure 13b) [148]. They have assembled complex light paths in a rectangular box and focused their focal point on a displacement stage so that a wide variety of items can be processed quickly and over large areas [149,150,151]. Furthermore, a wide variety of structures have been instrumented with this device. For example, a controlled anti-icing micro/nanosurface was prepared on the surface of the wing. In ice wind tunnel tests, the anti-icing/de-icing performance was quantitatively greater compared to the unmachined surface [152].

In addition, others have combined laser interference and hydrothermal methods to prepare graded micro/nanostructures for anti-icing surfaces [147,153,154]. Among them, Liu et al. [155] used a natural taro leaf model as an example and prepared biomimetic taro leaf structures on Ti6Al4V surfaces using this method. Laser interference produced a micropillar array structure (MPA) that reduced the contact area while creating a large number of cavities that could hinder the heat transfer effect. The hydrothermal treatment produced a large number of nanograss structures (NG) on the MPA that further reduced the contact area between the substrate and the liquid droplets, resulting in a higher resistance to ice. In addition, the hydrothermal treatment produced TiO_2_ with lower thermal conductivity than metals and had a low thermal conductivity (thermal conductivity: metals > oxides). As shown in Figure 14c, the different surface structures show different DTs. The layered composite structure induces more cavitation, resulting in a higher icing resistance; it is greatly enhanced by the icing delay time (DT), which is about 3732 s for the layered composite (HC) surface. In addition, the ice adhesion strength is reduced due to the presence of many cavities. In the experiments, the ice adhesion strength of the untreated surface was 458 kPa, while the ice adhesion strength of the surface with the hierarchical composite structure was 106 kPa.

#### 3.5.2. Friction Reduction, and Hardness Enhancement

For some items, prolonged friction can cause wear and even scrap. The topics of friction and hardness are always the same, the reduction of friction in some cases reduces the damage of wear, the higher hardness increases its durability, and both extend the life of the workpiece.

The presence of microweave arrays can increase the effective load carrying capacity compared to smooth surfaces, and the temporary storage of lubricating fluid on the weave surface can provide continuous lubrication in the absence of lubrication when the speed of the friction pair is high. The presence of the microloom can reduce the real contact area between materials to achieve the effect of friction reduction [156,157,158], and capture the abrasive particles generated during the friction process to reduce wear at the same time, significantly improving the wear problem caused by friction particles and optimizing the tribological performance between the friction interface. The specific model is shown in Figure 15. This technique uses the patterns (spatial multi-periodic energy distribution) generated by multi-beam laser interference to directly process or modify the material surface (curved or flat) to form high-precision nanoscale, microscale, and micro/nano-dual scale hybrid periodic structures corresponding to the patterns, obtaining modified materials with special chemical and physical properties [131].

For example, Wei et al. [130] combined laser interference manufacturing technology with patterning technology to provide a method of surface patterning for the preparation of low-friction, high-hardness artificial hip joints in terms of surface modification of artificial hip ball heads. Experimental results indicate that the sample surface modified with LIL presents better tribological performances and hardness properties than those untreated materials, including a 64% friction coefficient reduction and 40% hardness enhancement. This study provides a low-cost, high efficiency method for the design and fabrication of artificial hip joints with the improved tribological performances and hardness properties, which is very promising to significantly reduce the mean revision rate of post primary total hip replacements in the future. This method is an improvement of the existing artificial hip joint material, which greatly extends its service life.

## 4. Current Challenges and Prospects

With rapid development of science and technology, high speed, cheap, and efficient production methods are eagerly desired. LIL is one of the most promising technologies for processing micro- and nano-periodic structures with its high flexibility, large processing area, high controllability, and low cost. This work reviews the basic principles of LIL and describes the factors affecting the laser light field: spatial angle, angle of incidence, wavelength, polarization state, and the number of laser beams, and describes their relationship to the laser interference light field.

LIL has been developed for decades for the fabrication of micro-, sub-micro-, and nano-periodic structures, and has several applications in the fields of available structural color preparation, SERS, antireflective surfaces, bio-cells, and other industrial surface preparations, and this paper presents a review of typical applications over the years. Although these applications have many advantages, most of the research results are still in the experimental stage and have not yet been fully applied to industrial production. To industrialize and civilize laser interference lithography is a problem for which many researchers and scholars around the world are committed to solve, and it is also the direction of future development of laser interference lithography. At present, the problems of the industrialization of laser interference lithography are:High operating environment requirements: Due to the high number of optical components of the equipment, various optical components are sensitive to impurities in the surrounding environment and require a high-standard clean room; optical mirror frames are sensitive to vibration and require an optical platform for vibration filtering that otherwise affects stability;High equipment cost: The cost of laser interference lithography equipment is greater than the traditional lithography equipment (e.g., laser marking machine, etc.) because of the need to use lasers, optical anti-vibration platforms, displacement tables, the need for more optical lenses, and their fixing equipment; andRestricted material types: Due to the selectivity of materials to the laser wavelength, the laser with a certain wavelength is only applicable to certain materials, such as the wavelength of 1064 nm can process metals, silicon-based semiconductors, etc. Other materials, such as glasses, plastics, and other organic materials, have a poor etching effect, and the interferometric system needs to replace the UV laser and corresponding optical components.

These are only some of the stumbling blocks already known in the industrialization of laser interference lithography. In recent years, many researchers have been trying to overcome the problems. To solve the problem of high environmental requirements, researchers from the Fraunhofer IWS have combined interferometric lithography systems in a rectangular box with high integration, avoiding the overwhelming influence of dust and vibrations on lenses, and improving stability performance [18,159]. As for the issue of high equipment cost, the price of lasers and some optoelectronic components is decreasing every year as the technology develops, so the cost of the system should be significantly lower in the future. Concerning the problem of restricted material types, a solution has been proposed to integrate several lasers of various wavelengths in a single laser interference system [18].

In the future, with the continuous efforts of researchers, LIL technology, as a non-contact processing technology with great potential, will overcome all obstacles and be applied in more industrial fields.

## Figures and Tables

**Figure 1 nanomaterials-13-01818-f001:**
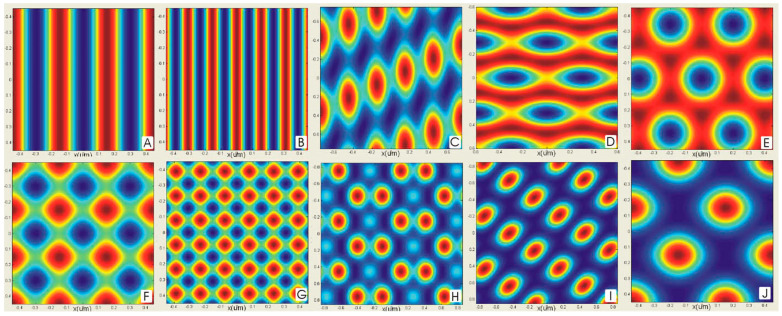
MATLAB simulations of interfering light fields: (**A**,**B**) stripe arrays obtained from double-beam LIL at different incidence angles. (**C**–**E**) Dot/hole arrays obtained from triple-beam LIL at different azimuthal angles. (**F**,**G**) Dot/hole arrays obtained from quadruple-beam LIL at different incidence angles. (**H**–**J**) Dot/hole arrays produced by quadruple-beam LIL at different azimuthal angles. Reproduced with permission/adapted from [16].

**Figure 2 nanomaterials-13-01818-f002:**
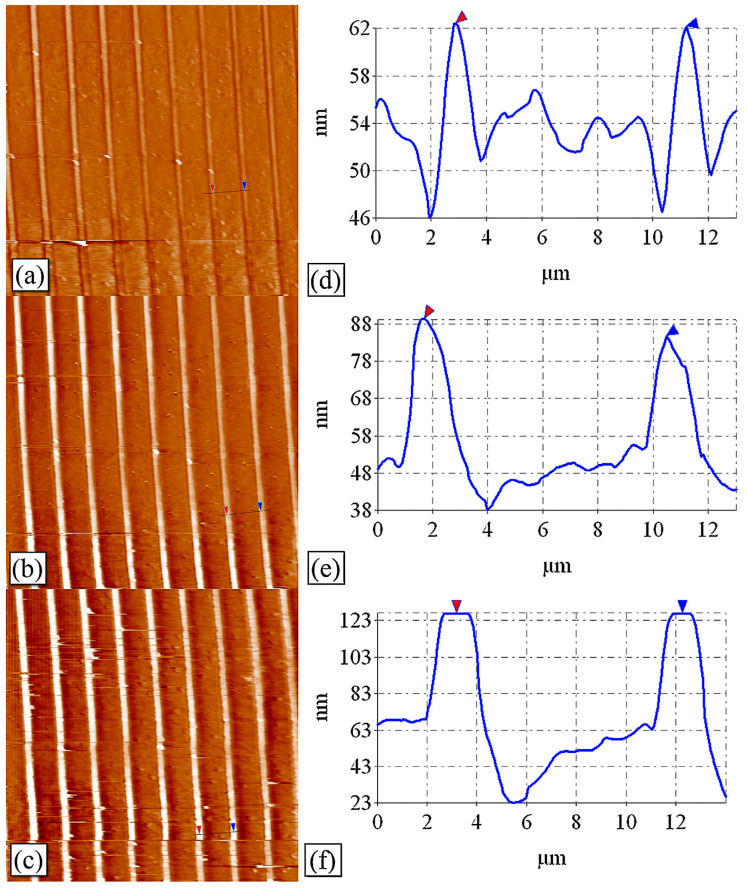
AFM images of a single-pulse double-beam LIL: (**a**–**c**) for exposure energies of 560 mJ·cm^−2^, 630 mJ·cm^−2^, and 700 mJ·cm^−2^; (**d**–**f**) are views of the cross-sectional wheelhouse for (**a**), (**b**), and (**c**), respectively. Reproduced with permission/adapted from [25].

**Figure 3 nanomaterials-13-01818-f003:**
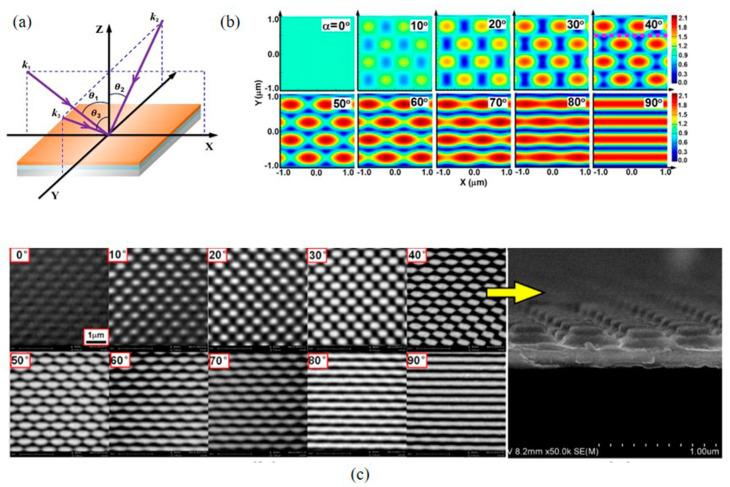
(**a**) Schematic illustration of the interference geometry. (**b**) Simulation results of the interference patterns using the geometry in (**a**) with the polarization direction of the input laser beam changed from α = 0 to 90 degrees with respect to the vertical direction. (**c**) Experimental results of the produced photoresist grating structures. Cross-section demonstration of the fabricated structures with α = 40°, where the sample was cut along the dashed pink line in (**b**). Reproduced with permission/adapted from [29].

**Figure 4 nanomaterials-13-01818-f004:**
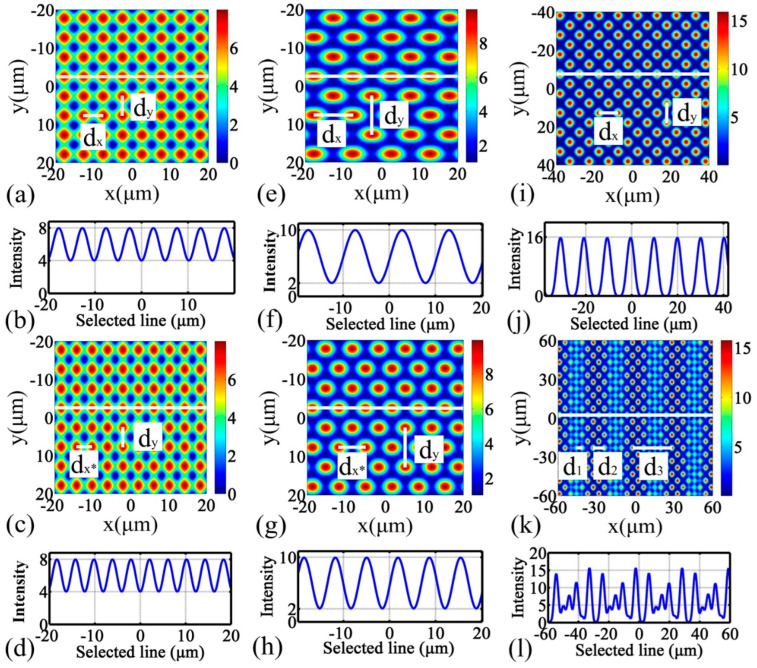
MATLAB simulations of LIL for three different polarization modes: The TE-TE-TE-TE mode includes figure (**a**) with the same angle of incidence and figure (**c**) with a misaligned angle of incidence. The TE-TE-TE-TM mode includes figure (**e**) with the same angle of incidence and figure (**g**) with a mis-aligned angle of incidence. The TE-TE-TM-TM pattern includes figure (**i**) with the same angle of incidence and figure (**k**) with a misaligned angle of incidence. (**b**,**d**,**f**,**h**,**j**,**l**) are the intensity profiles along the double-arrow lines in (**a**,**c**,**e**,**g**,**i**,**k**), respectively. Reproduced with permission/adapted from [26].

**Figure 5 nanomaterials-13-01818-f005:**
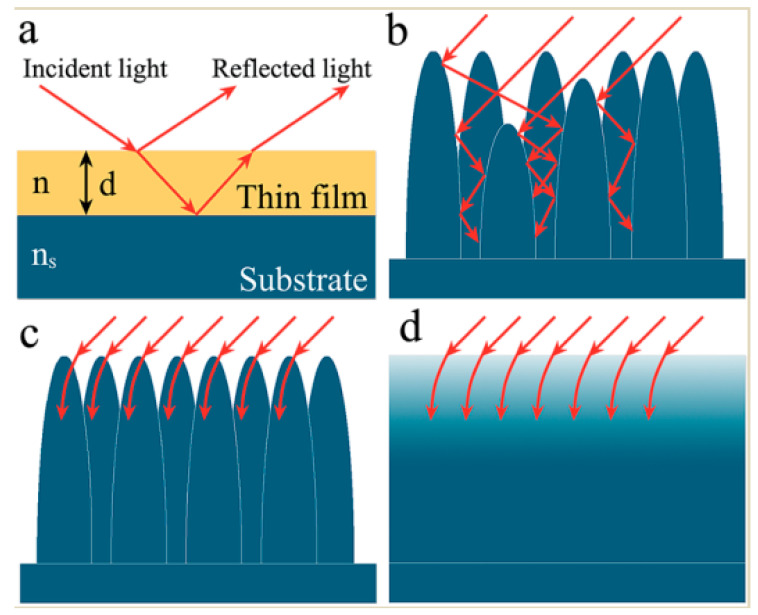
Schematic diagram of the structure with reduced reflection of the incident light. (**a**) Propagation of incident light through a single layer film on a substrate (n_s_ > n). (**b**) Multiple internal reflections of incident light in a microstructure array. (**c**) Interaction of incident light with the subwavelength-size nanoarray. (**d**) Schematic illustration of the refractive index change corresponding to (**c**). Reproduced with permission/adapted from [39].

**Figure 6 nanomaterials-13-01818-f006:**
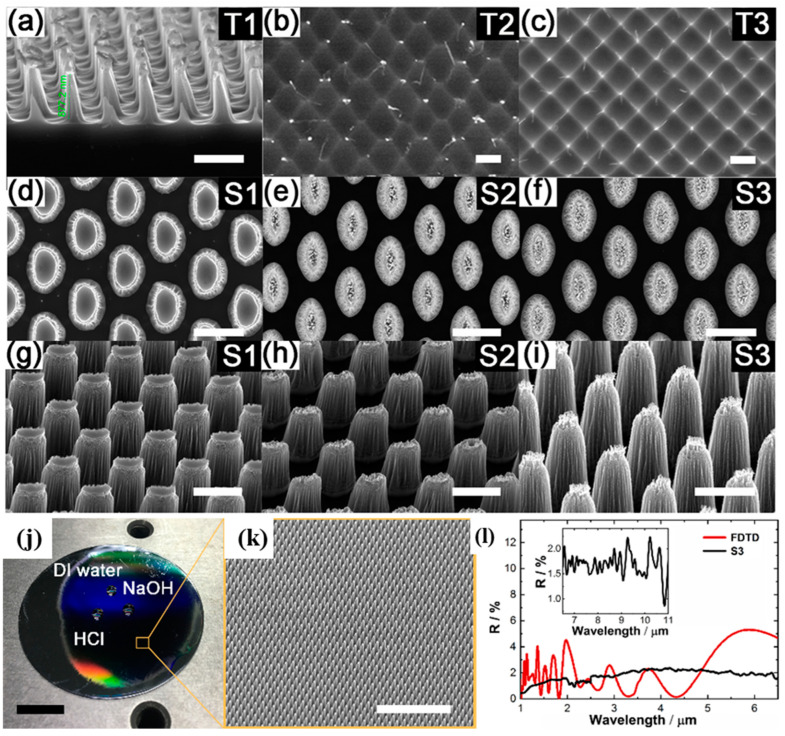
(**a**–**f**) Cross-sectional and top-view SEM images of the samples. (**g**–**i**) SEM images of samples (S1, S2, and S3) placed at 45°, 30°, and 45°, respectively. Scale bar: 1 μm. (**j**) Photograph of the DI water, HCl, and NaOH droplets on sample S3 (scale bar: 1 cm). (**k**) SEM image corresponding to sample S3 (scale bar: 10 μm). (**l**) The simulated (red curve) and experimental (black curve) reflectance spectra of the hierarchical moth-eye structure. The inset is the reflectance spectrum in the mid-infrared range. Reproduced with permission/adapted from [45].

**Figure 7 nanomaterials-13-01818-f007:**
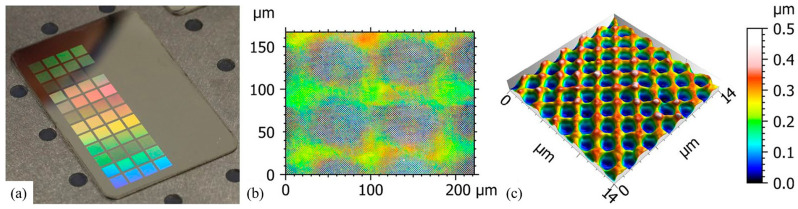
(**a**) Photograph of the stainless steel surface treated with different numbers of pulses (from 1 to 5) and laser fluences (from 0.7 to 11.4 J/cm^2^); (**b**) confocal image of LIL holographic pixels on the steel surface with a diameter of approximately 50 µm; (**c**) typical hole-like pattern with a spatial period of 1.8 µm and a structure depth of 0.3 µm (the used laser fluence was 1.9 J/cm^2^ and three pulses were applied). Reproduced with permission/adapted from [47].

**Figure 8 nanomaterials-13-01818-f008:**
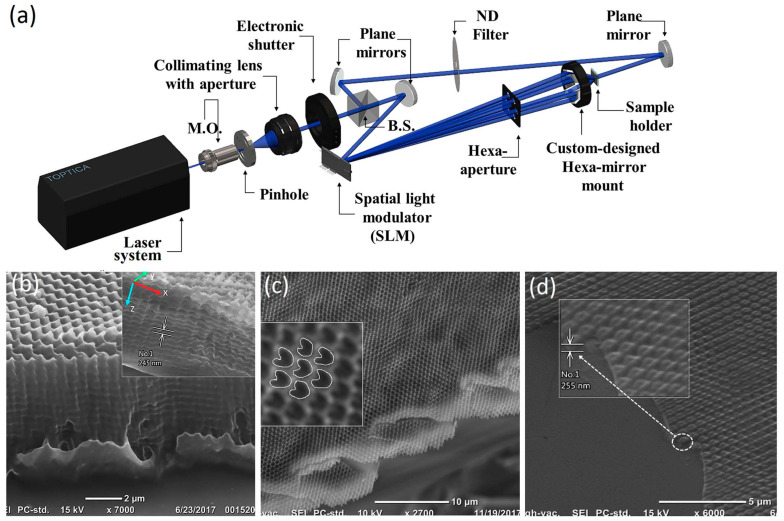
(**a**) An experimental setup to realize the helical photonic crystals with a submicron scale spatial and axial periodicity; 50:50 ultraviolet beam splitter for 405 nm; the microscopic objective of 10×; SEM images of the fabricated submicron periodic helical photonic crystal structure on positive photoresist (AZ 1518). (**b**) Cross-sectional 45° tilt view presenting the realization of 3D periodic structures over a large area; the inset shows another sample with different exposures in the isometric view obtained through tilt and rotation of the sample stage. (**c**) Cross-sectional view of a purposely broken sample with less exposure time. (The inset shows a magnified view with chirality properly marked). (**d**) A 45° tilt view of a single-layer helical structure over a single pitch in negative photoresist (marked by dotted circle). Reproduced with permission/adapted from [41].

**Figure 9 nanomaterials-13-01818-f009:**
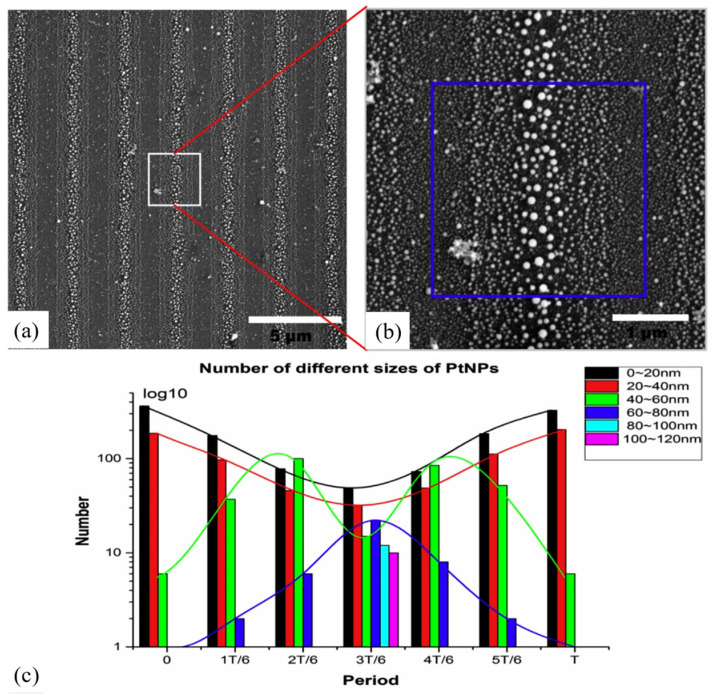
(**a**,**b**) SEM images of periodic variable-sized platinum nanoparticles (PtNPs) fabricated by double-beam LIL and (**c**) the distribution bar of different sizes of platinum nanoparticles (PtNPs) within an area of T × T along the period, where T is the period. Reproduced with permission/adapted from [33].

**Figure 10 nanomaterials-13-01818-f010:**
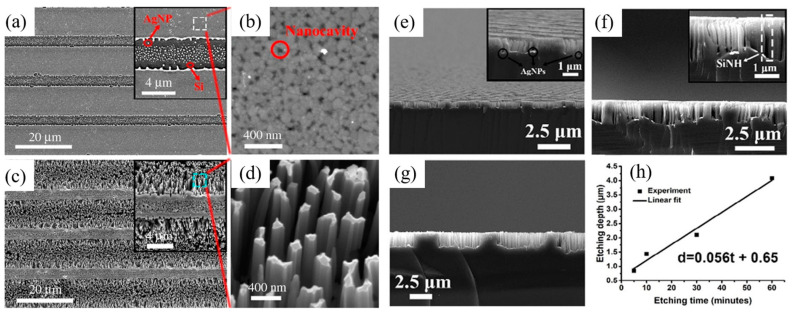
(**a**) Ordered Ag patterns obtained by SELIL. (**b**) Ag nanocavities at high magnification. (**c**) Corresponding hybrid SiNW and SiNH arrays after MACE for 30 min, and (**d**) SiNWs at high magnification. (**e**–**g**) SEM cross-sectional images of hybrid SiNW and SiNH arrays etched for (**e**) 5, (**f**) 10, and (**g**) 30 min; and (**h**) the relationship between the etching depth and etching time. Reproduced with permission/adapted from [100].

**Figure 11 nanomaterials-13-01818-f011:**
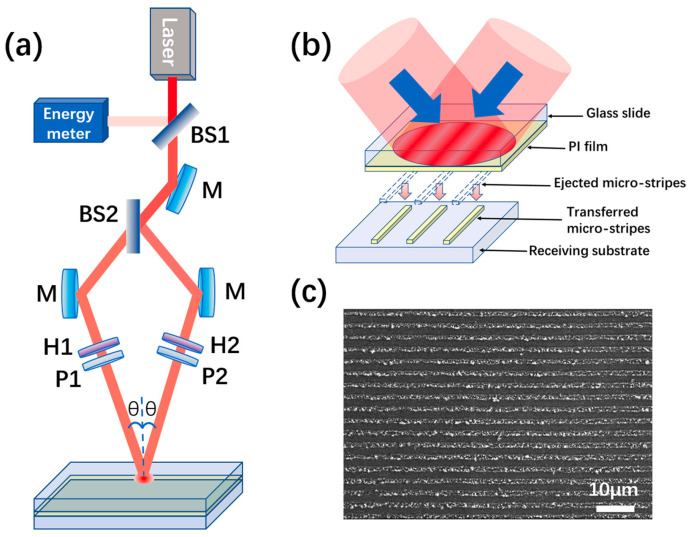
Experimental setup and formation schematic of the laser interference-induced forward transfer PI micro-stripes. (**a**) Schematic diagram of the double-beam LIIFT configuration. (**b**) The fabrication schematic of micro-stripes in LIIFT, and the micro-stripes eject from the PI-donor film to the receiving substrate. (**c**) The transferred micro-stripes from the donor-film 1.2 µm with the laser fluence 39 mJ·cm^−2^ and pulse number 50. Reproduced with permission/adapted from [105].

**Figure 12 nanomaterials-13-01818-f012:**
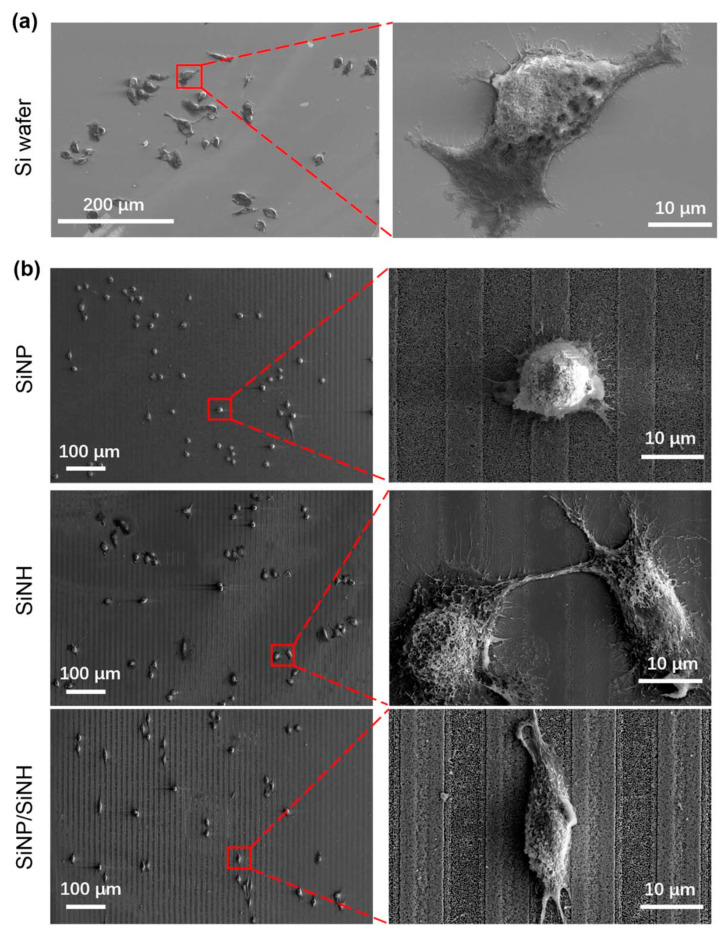
(**a**) SEM images of A549 cells cultured on the silicon wafer for 24 h. The image on the right side shows the details of cells in the red box with a high magnification. (**b**) SEM images of A549 cells cultured on the silicon nanopillar (SiNP) arrays, SiNH, and SiNP/SiNH arrays for 24 h. The images on the right side show the details of cells in the red boxes with a high magnification. Reproduced with permission/adapted from [114].

**Figure 13 nanomaterials-13-01818-f013:**
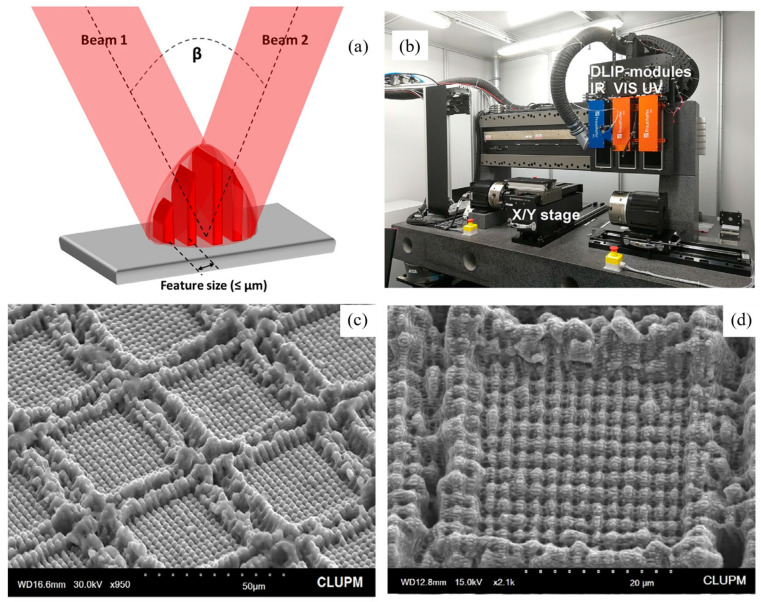
(**a**) Illustration of the direct laser interference patterning process for double-beam LIL and (**b**) the experimental setup of the LIL system with a ps laser source. Reproduced with permission/adapted from [18]. (**c**,**d**) SEM images of samples patterned with hierarchical microstructures, via LIL and direct laser writing techniques. Reproduced with permission/adapted from [151].

**Figure 14 nanomaterials-13-01818-f014:**
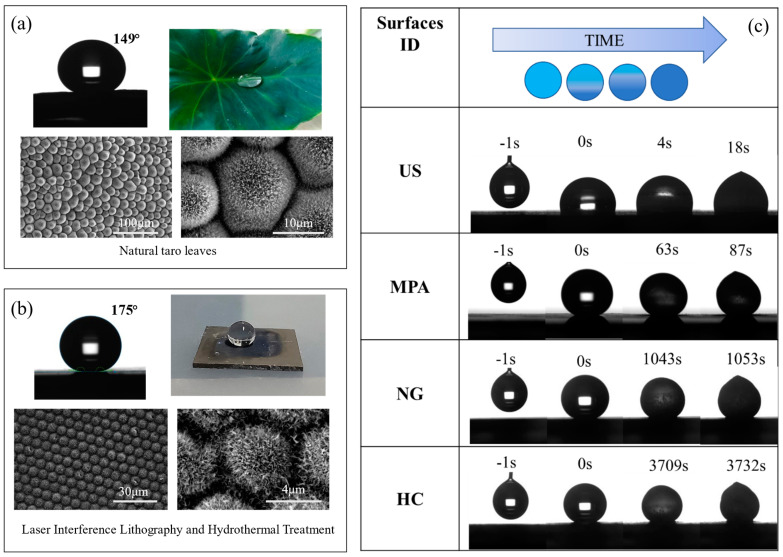
(**a**) Images of the contact angle and the shape of water droplets on a natural taro leaf surface; SEM images of the taro leaf surface structures. (**b**) Images of the contact angle and the shape of water droplets on the processed HC surface; SEM images of the processed HC surfaces. The microstructures consisted of the micropillar array with the period of 10 μm; The nanostructure array covered the entire microstructure surface, and the nanostructure was composed of TiO_2_ nanoscale grasses with the diameter of −40 nm and height of −1.5 μm. (**c**) The photos of the untreated surfaces (US)-, MPA-, NG-, and HC- surfaces (from the top to bottom). The reference droplets (4 μL tap water) were placed on the four surfaces at −10 °C. All of the droplets were initially transparent. Following the DT of 18 s, the droplet became opaque on the US surface; after 87 s, another droplet was opaque on the MPA surface. Following 1053 s of DT, the droplets also became opaque on the NG surface. Prior to the time of 3709 s, the droplets on the HC surface were still transparent. For the next 23 s, the droplets gradually froze and the transmittance decreased. At 3732 s, the droplets became opaque. Reproduced with permission/adapted from [155].

**Figure 15 nanomaterials-13-01818-f015:**
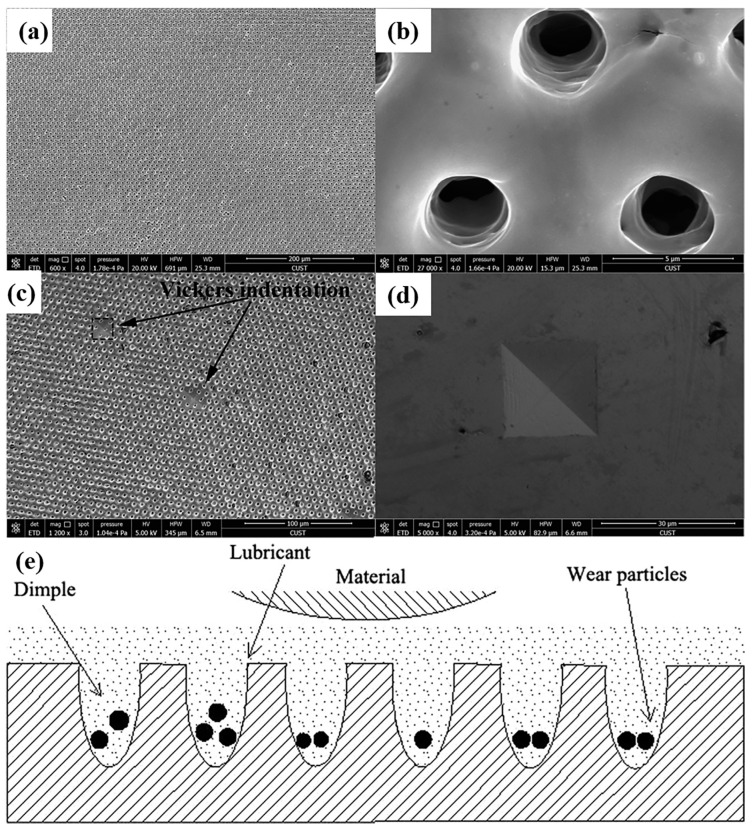
(**a**) SEM image of the microstructure fabricated by triple-beam LIL; (**b**) close-up image of the structured surface; (**c**) SEM image of Vickers’ indentation with the applied load of 200 g for 10 s; (**d**) SEM image of the original material with Vickers’ indentation. (**e**) Schematic of the lubrication model reducing friction by the formation of dimples. Reproduced with permission/adapted from [130].

## Data Availability

Not applicable.

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
