# Peer review of "Laser Interference Lithography—A Method for the Fabrication of Controlled Periodic Structures"

_nanomaterials, 2023, doi:10.3390/nano13121818_

Round 1
Reviewer 1 Report
This review article described the LIL fabrication technology and its application in functional surfaces for antireflection, structural colors, SERS, friction reduction, biological cell modulation, etc. Then, the authors mentioned challenges and problems regarding the LIL technology in general perspective. I think this review concerning the LIL technique is helpful for researchers working in related fields. Although they treated versatile applications of the LIL with many literatures, they didn’t well explain the principle of the LIL techniques in some parts. Therefore, I think the current form of this manuscript is not suitable for publication in Nanomaterials. The quality of the manuscript should be further improved by addressing the following points.
1. The author mentioned that they described the principles of LIL techniques in terms of beam number, spatial angle, angle of incidence, wavelength, and polarization of incident light. Some of them were adequately explained, but others were not well explained. That is, the author only mentioned the results without briefly explaining the basic principles involved in the sentences below.
Line 165-168 : In three-beam interference at equal angles, the modulation mode is not formed. In four-beam interference, the azimuth changes with the modulation period. In the TE-TM-TE-TM polarization mode, the modulation period increases as the azimuth angle decreases.
Line 194-198 : Figure 3(b) shows the simulation results of the interference pattern within the 22 μm2 region, and Figure 3(c) shows the experimental results of the corresponding photoresist grating with an area of 55 μm2. It can be seen that changing the different polarization states can directly affect the light field energy distribution of the interference.
2. In line 118, the authors mentioned that “Figure 1 shows the three-beam DLIL setup and the computer simulation results”. However, there is no three-beam DLIL setup in Figure 1.
3. The explanation of the simulation results in Figure 1 is repeated on several pages (bottom of page 2, middle of page 3, and bottom of page 3). It is necessary to clearly explain the simulation results in Figure 1 in one place.
4. The author mentioned the issues for commercialization of LIL technology such as high operating environmental requirements, high equipment cost, and restricted material types. In addition, it is required to achieve reliable pattern uniformity over large areas. I think the authors need to describe recent research trends to overcome these issues for the readers working in related fields.
There are some errors in English grammar. The manuscript should be carefully proofread by a native professional. Below are just two example sentences to be corrected.
Line 6, page 5 : Polarized state as the intensity condition of the laser pattern profile.
Line 9-13, page 5 : Jianfang He made the relevant research analysis[29]. By placing a half-wave plate at the input coupling or in front of the binding end of the fiber. The polarization direction of the initial vertically polarized 325 nm laser beam was varied from α = 0 to 90 degrees in steps of 10 degrees. Where α is the angle of the in-plane polarization direction relative to the vertical axis perpendicular to the axis of the fiber beam.
Author Response
Dear Editor/ Reviewers,
Thank you for all the comments. The following are our responses.
Reviewer #1
This review article described the LIL fabrication technology and its application in functional surfaces for antireflection, structural colors, SERS, friction reduction, biological cell modulation, etc. Then, the authors mentioned challenges and problems regarding the LIL technology in general perspective. I think this review concerning the LIL technique is helpful for researchers working in related fields. Although they treated versatile applications of the LIL with many literatures, they didn’t well explain the principle of the LIL techniques in some parts. Therefore, I think the current form of this manuscript is not suitable for publication in Nanomaterials. The quality of the manuscript should be further improved by addressing the following points.
Comment 1. The author mentioned that they described the principles of LIL techniques in terms of beam number, spatial angle, angle of incidence, wavelength, and polarization of incident light. Some of them were adequately explained, but others were not well explained. That is, the author only mentioned the results without briefly explaining the basic principles involved in the sentences below.
Line 165-168 : In three-beam interference at equal angles, the modulation mode is not formed. In four-beam interference, the azimuth changes with the modulation period. In the TE-TM-TE-TM polarization mode, the modulation period increases as the azimuth angle decreases.
Line 194-198 : Figure 3(b) shows the simulation results of the interference pattern within the 22 μm2 region, and Figure 3(c) shows the experimental results of the corresponding photoresist grating with an area of 55 μm2. It can be seen that changing the different polarization states can directly affect the light field energy distribution of the interference.
Response:
As suggested, the relevant descriptions/explanations have been added to the manuscript and the changes are marked in yellow.
Comment 2. In line 118, the authors mentioned that “Figure 1 shows the three-beam DLIL setup and the computer simulation results”. However, there is no three-beam DLIL setup in Figure 1.
Response:
As suggested, this issue has been corrected in the manuscript and the changes are marked in yellow.
Comment 3. The explanation of the simulation results in Figure 1 is repeated on several pages (bottom of page 2, middle of page 3, and bottom of page 3). It is necessary to clearly explain the simulation results in Figure 1 in one place.
Response:
As suggested, this issue has been corrected in the manuscript and the changes are marked in yellow.
Comment 4. The author mentioned the issues for commercialization of LIL technology such as high operating environmental requirements, high equipment cost, and restricted material types. In addition, it is required to achieve reliable pattern uniformity over large areas. I think the authors need to describe recent research trends to overcome these issues for the readers working in related fields.
Response:
As suggested, the relevant description of this issue has been added to the '4. Current Challenges and Prospects' section of the manuscript, and the changes are highlighted in yellow.
Comments on the Quality of English Language
There are some errors in English grammar. The manuscript should be carefully proofread by a native professional. Below are just two example sentences to be corrected.
Line 6, page 5 : Polarized state as the intensity condition of the laser pattern profile.
Line 9-13, page 5 : Jianfang He made the relevant research analysis[29]. By placing a half-wave plate at the input coupling or in front of the binding end of the fiber. The polarization direction of the initial vertically polarized 325 nm laser beam was varied from α = 0 to 90 degrees in steps of 10 degrees. Where α is the angle of the in-plane polarization direction relative to the vertical axis perpendicular to the axis of the fiber beam.
Response:
As suggested, numerous language/English usage errors were corrected and significant changes were made to the revised draft.
Thank you so much for your double check. All changes are highlighted in color in the revised manuscript. We would like to thank the referees again for taking the time to review our manuscript.
Thanks for your work/help.
sincerely
Corresponding Author:
Wang Zuobin, PhD (Warwick)
Changbai Scholars Professor/Director
China International Research Center for Nanoprocessing and Manufacturing
Changchun University of Science and Technology
No. 7089 Weixing Road, Changchun 130022, China
Email: WangZ@cust.edu.cn
Tel: +86 431 85582341
http://www.3m-nano.org
http://cnm.cust.edu.cn/en
Founding Chairman of IEEE 3M-NANO
Reviewer 2 Report
This work consists in a review of recent (and less recent) developments of LIL.
Despite the amount of available information, Authors made a considerable effort to organize it in a logical way, and this has resulted in a manuscript of a good level.
Since this is a review, its figures have been taken from the original papers and Authors have chosen to acknowledge this fact by introducing the corresponding reference in the figure caption. However, this makes it not clear whether the figure is simply a reproduction or has been adapted from the original reference. For instance, in the case of Fig. 1, this figure has been simply reproduced, and, as such, written permission should have been granted by the Publisher. I think the best way to solve this ambiguity consists in adding at the end of each caption a sentence like: “Reproduced with permission/Adapted from Ref. [xyz]”
Moreover, although the general structure of the manuscript is clear, there a few parts requiring improvements, often revealing simply the lack of a complete final check for self-consistency between figures, cations and main text.
Here below is a list of the points requiring improvements:
Line 105: “Micro-nano structures with different feature sizes are achieved using DLIL. “ No DLIL acronym has not been defined so far. Please define (or correct).
Line 118: “Figure. 1 shows the three-beam DLIL setup” No DLIL acronym has not been defined so far. Please define (or correct).
Line 187: “Line Polarized state as the intensity condition of the laser pattern profile”. This sentence is supposed to be a statement, not a title. Therefore, a verb is missing, which makes it difficult to understand its meaning. Please reformulate.
Line 190-191:” By placing a half-wave plate at the input coupling or in front of the binding end of the fiber”. This sentence can be better understood if it is unified with the previous one.
Line 193-194: “of 10 degrees. Where α is the angle of the in-plane polarization direction relative to the vertical axis perpendicular to the axis of the fiber beam.” sound better if modified as: “of 10 degrees, where α is the angle of the in-plane polarization direction relative to the vertical axis and perpendicular to the axis of the fiber beam.”
Lines 238-240: “Among the wide range of renewable energy sources available, 238 solar photovoltaic technology is considered to be the cleanest and safest of the large-scale power generation technologies[34,35].” This statement is quite questionable. Authors should consider that large-scale wind generator (i.e. the ones rated over 10 MW) are considered to be the technology with the lowest carbon footprint, with a degree of safety comparable to the one of solar technology. A reformulation of this statement is suggested.
Lines 257-259: “How can a solar panel of the same material increase the efficiency of a solar panel? People think of is to leave more solar energy!” I’m not able to grasp the meaning of these sentences, especially the second one. The same will probably apply to other readers. Please consider to reformulate in a clearer way.
Lines 297-299: “The broadband antireflective multifunctional micro-nano composite structure with superhydrophobic and anti-corrosion properties has been realized.” This sentence is just an unnecessary repetition of the previous one. Authors should remove it.
Line 335: “DLIP”. This acronym has not been defined. Please define it.
Lines 383-384: “surface enhanced spectroscopy (SERS)”. Please correct to “surface enhanced Raman spectroscopy (SERS)”
Line 416: “LIP“. This acronym is not defined. Authors probably mean “LIL”. Please correct.
Lines 429-430: “superluminescence-enhanced laser interference lithography”. The definition of the corresponding acronym is missing. I suggest to modify as: “superluminescence-enhanced laser interference lithography (SELIL)”.
Line 433: “Ag molecules”. The term “molecules” sounds very odd when applied to materials with metallic bonds. I suggest to modify it as “Ag particles”.
Line 458: “laser interference forward transfer”. Please modify as “laser interference induced forward transfer”.
Lines 499-500” : “Xiaomin Wu et al.[113-115] based on the preparation of micro- nano structures of different morphologies of LIL and” . This sentence is not clear in its meaning. Authors probably meant “Xiaomin Wu et al.[113-115] based their work on the preparation of micro- nano structures of different morphologies of LIL and”. Please clarify and reformulate.
Line 554: “DLIP”. This acronym has not been defined. Please define it.
Lines 580-581: “Ice formation on the HC-, NG-, MPA-, 580 and US- surfaces at -10 °C with different DTs.” This sentence in the caption of Figure 14 is presently referring to panel (b), which is incorrect. Please amend it.
Line 581, 584, 590: The US- surface has not been defined in the main text, so no explanation is available to reader. Please include a description there.
Lines 587-590: “(b) Comparison of the DT of ice formation at temperatures below -10 °C. The DT of HC surface was 3732 s (see the last column), the DT of NG surface was 1053 s (see the third column), the DT of MPA surface was 87 s (see the second column), and the DT of US surface was 18 s (see the first column)”. Panel b has already been correctly described in the caption. This description probably refers to another panel to be labelled as “d”, but this panel is missing in the Figure. Please complete the figure.
Line 623: “(e) The whole indentation pattern spot.” Panel (e) is missing in Figure 15. Please correct.
After correction of these shortcoming, I think the paper can be published.
General level is good. Some sentences that cannot be clearly understood have already been highlighted in the previous section.
Author Response
Dear Editor/ Reviewers,
Thank you for all the comments. The following are our responses.
Reviewer #2
This work consists in a review of recent (and less recent) developments of LIL.
Despite the amount of available information, Authors made a considerable effort to organize it in a logical way, and this has resulted in a manuscript of a good level.
Since this is a review, its figures have been taken from the original papers and Authors have chosen to acknowledge this fact by introducing the corresponding reference in the figure caption. However, this makes it not clear whether the figure is simply a reproduction or has been adapted from the original reference. For instance, in the case of Fig. 1, this figure has been simply reproduced, and, as such, written permission should have been granted by the Publisher. I think the best way to solve this ambiguity consists in adding at the end of each caption a sentence like: “Reproduced with permission/Adapted from Ref. [xyz]”
Moreover, although the general structure of the manuscript is clear, there a few parts requiring improvements, often revealing simply the lack of a complete final check for self-consistency between figures, cations and main text.
Here below is a list of the points requiring improvements:
Line 105: “Micro-nano structures with different feature sizes are achieved using DLIL. “ No DLIL acronym has not been defined so far. Please define (or correct).
Line 118: “Figure. 1 shows the three-beam DLIL setup” No DLIL acronym has not been defined so far. Please define (or correct).
Line 187: “Line Polarized state as the intensity condition of the laser pattern profile”. This sentence is supposed to be a statement, not a title. Therefore, a verb is missing, which makes it difficult to understand its meaning. Please reformulate.
Line 190-191:” By placing a half-wave plate at the input coupling or in front of the binding end of the fiber”. This sentence can be better understood if it is unified with the previous one.
Line 193-194: “of 10 degrees. Where α is the angle of the in-plane polarization direction relative to the vertical axis perpendicular to the axis of the fiber beam.” sound better if modified as: “of 10 degrees, where α is the angle of the in-plane polarization direction relative to the vertical axis and perpendicular to the axis of the fiber beam.”
Lines 238-240: “Among the wide range of renewable energy sources available, 238 solar photovoltaic technology is considered to be the cleanest and safest of the large-scale power generation technologies[34,35].” This statement is quite questionable. Authors should consider that large-scale wind generator (i.e. the ones rated over 10 MW) are considered to be the technology with the lowest carbon footprint, with a degree of safety comparable to the one of solar technology. A reformulation of this statement is suggested.
Lines 257-259: “How can a solar panel of the same material increase the efficiency of a solar panel? People think of is to leave more solar energy!” I’m not able to grasp the meaning of these sentences, especially the second one. The same will probably apply to other readers. Please consider to reformulate in a clearer way.
Lines 297-299: “The broadband antireflective multifunctional micro-nano composite structure with superhydrophobic and anti-corrosion properties has been realized.” This sentence is just an unnecessary repetition of the previous one. Authors should remove it.
Line 335: “DLIP”. This acronym has not been defined. Please define it.
Lines 383-384: “surface enhanced spectroscopy (SERS)”. Please correct to “surface enhanced Raman spectroscopy (SERS)”
Line 416: “LIP“. This acronym is not defined. Authors probably mean “LIL”. Please correct.
Lines 429-430: “superluminescence-enhanced laser interference lithography”. The definition of the corresponding acronym is missing. I suggest to modify as: “superluminescence-enhanced laser interference lithography (SELIL)”.
Line 433: “Ag molecules”. The term “molecules” sounds very odd when applied to materials with metallic bonds. I suggest to modify it as “Ag particles”.
Line 458: “laser interference forward transfer”. Please modify as “laser interference induced forward transfer”.
Lines 499-500” : “Xiaomin Wu et al.[113-115] based on the preparation of micro- nano structures of different morphologies of LIL and” . This sentence is not clear in its meaning. Authors probably meant “Xiaomin Wu et al.[113-115] based their work on the preparation of micro- nano structures of different morphologies of LIL and”. Please clarify and reformulate.
Line 554: “DLIP”. This acronym has not been defined. Please define it.
Lines 580-581: “Ice formation on the HC-, NG-, MPA-, 580 and US- surfaces at -10 °C with different DTs.” This sentence in the caption of Figure 14 is presently referring to panel (b), which is incorrect. Please amend it.
Line 581, 584, 590: The US- surface has not been defined in the main text, so no explanation is available to reader. Please include a description there.
Lines 587-590: “(b) Comparison of the DT of ice formation at temperatures below -10 °C. The DT of HC surface was 3732 s (see the last column), the DT of NG surface was 1053 s (see the third column), the DT of MPA surface was 87 s (see the second column), and the DT of US surface was 18 s (see the first column)”. Panel b has already been correctly described in the caption. This description probably refers to another panel to be labelled as “d”, but this panel is missing in the Figure. Please complete the figure.
Line 623: “(e) The whole indentation pattern spot.” Panel (e) is missing in Figure 15. Please correct.
After correction of these shortcoming, I think the paper can be published.
Response:
Thank you so much for your careful check. All the changes are highlighted with color in the revised manuscript. We would like to thank the referee again for taking the time to review our manuscript.
Thank you for your work / help.
Sincerely,
Corresponding author:
Zuobin Wang, PhD (Warwick)
Changbai Scholar Professor / Director
International Research Centre for Nano Handling and Manufacturing of China
Changchun University of Science and Technology
7089 Weixing Road, Changchun 130022, China
Email: WangZ@cust.edu.cn
Phone: +86 431 85582341
http://www.3m-nano.org
http://cnm.cust.edu.cn/en
IEEE 3M-NANO Founding Chair
Reviewer 3 Report
The manuscript “Laser Interference Lithography ˗ A Method for Fabrication of 2 Controlled Periodic Structures” described by Ri Liu, Liang Cao, Dongdong Liu, Lu Wang, Sadaf Saeed and Zuobin Wang presents an interesting analysis of scientific publications on the possibility of using the LIL technique in the fabrication of high-resolution periodic structures and may be accepted for publication in Journal
Author Response
Dear Editor/ Reviewers,
Thank you for all the comments. The following are our responses.
Reviewer #3
The manuscript “Laser Interference Lithography ˗ A Method for Fabrication of 2 Controlled Periodic Structures” described by Ri Liu, Liang Cao, Dongdong Liu, Lu Wang, Sadaf Saeed and Zuobin Wang presents an interesting analysis of scientific publications on the possibility of using the LIL technique in the fabrication of high-resolution periodic structures and may be accepted for publication in Journal.
Response:
Thanks for the positive comments.
Thank you for your work / help.
Sincerely,
Corresponding author:
Zuobin Wang, PhD (Warwick)
Changbai Scholar Professor / Director
International Research Centre for Nano Handling and Manufacturing of China
Changchun University of Science and Technology
7089 Weixing Road, Changchun 130022, China
电子邮件: WangZ@cust.edu.cn
电话: +86 431 85582341
http://www.3m-nano.org
http://cnm.cust.edu.cn/en
IEEE 3M-NANO 创始主席
Round 2
Reviewer 1 Report
The authors appropriately corrected and reflected the points raised by the reviewer in the revised manuscript. Therefore, I recommend that this manuscript be published in Nanomaterial.